# INVESTIGATING MEMORIZATION IN VIDEO DIFFUSION MODELS

## ABSTRACT

Diffusion models, widely used for image and video generation, face a significant limitation: the risk of memorizing and reproducing training data during inference, potentially generating unauthorized copyrighted content. While prior research has focused on image diffusion models (IDMs), video diffusion models (VDMs) remain underexplored. To address this, we introduce new metrics specifically designed to separately assess content and motion memorization in VDMs. By applying these metrics, we systematically analyze memorization in various pretrained VDMs, including text-conditional and unconditional models on various datasets, revealing that memorization is widespread across both video and image datasets. Finally, we propose effective detection strategies for both content and motion memorization, offering a foundational approach for improving privacy in VDMs.

## 1 INTRODUCTION

Following the surge in popularity of image diffusion models (IDMs), which surpassed GANs (Goodfellow et al., 2014) in image synthesis quality (Nichol & Dhariwal, 2021), video diffusion models (VDMs) (OpenAI, 2024; Runway, 2024) have emerged as a significant advancement and attracted a lot of attention in both the research community and public attention due to their ability to generate novel and realistic videos. However, a critical limitation has been identified in IDMs that raises questions about the novelty of their outputs, where training images can be extracted by a pretrained IDM and regurgitated during generation (Carlini et al., 2023). Such memorized outputs have been criticized as "digital forgery"(Somepalli et al., 2023a), and several lawsuits(Saveri & Matthew, 2023) have been filed against companies for generating copyrighted artworks with these models.

Significant research has been conducted to understand and mitigate this memorization issue, identifying causes (Somepalli et al., 2023b) and proposing remedies (Daras et al., 2023; Wen et al., 2024; Chen et al., 2024a; Ren et al., 2024). These efforts have achieved notable success in IDMs. However, there is a lack of thorough investigation into memorization in VDMs, particularly in the influential text-conditional (T2V) models. The investigation of memorization in VDMs is arguably even more critical than in IDMs, as videos consist of multiple temporally consistent frames. This adds an additional dimension to consider, and alongside content memorization, motion memorization also needs to be addressed. Motion patterns are highly distinctive, much like biometrics, and their memorization raises significant privacy concerns. Previous work (Kale et al., 2004; Klempous, 2009; Nixon et al., 2010) has shown that unique physical traits, such as a person's gait, are sufficient for identification. Additionally, recent studies (Nair et al., 2023a;b) demonstrate that motion data can be used to identify individuals in virtual reality (VR) environments. As a result, addressing motion memorization is just as crucial as tackling content memorization, especially given the growing use of VDMs and their broad range of downstream applications. A recent study (Rahman et al., 2024) attempted to explore memorization in VDMs. However, its definitions and evaluations of both content and motion memorization are narrow in scope and lack broad applicability, which has hindered its wider adoption in the research community.

To address these limitations, we have made the following key contributions:

1. **Redefinition of content and motion memorization** (Sec. 2). We propose more practical definitions for both content and motion memorization with a focus on privacy preservation. Specifically, we redefine content memorization as a frame-level phenomenon independent

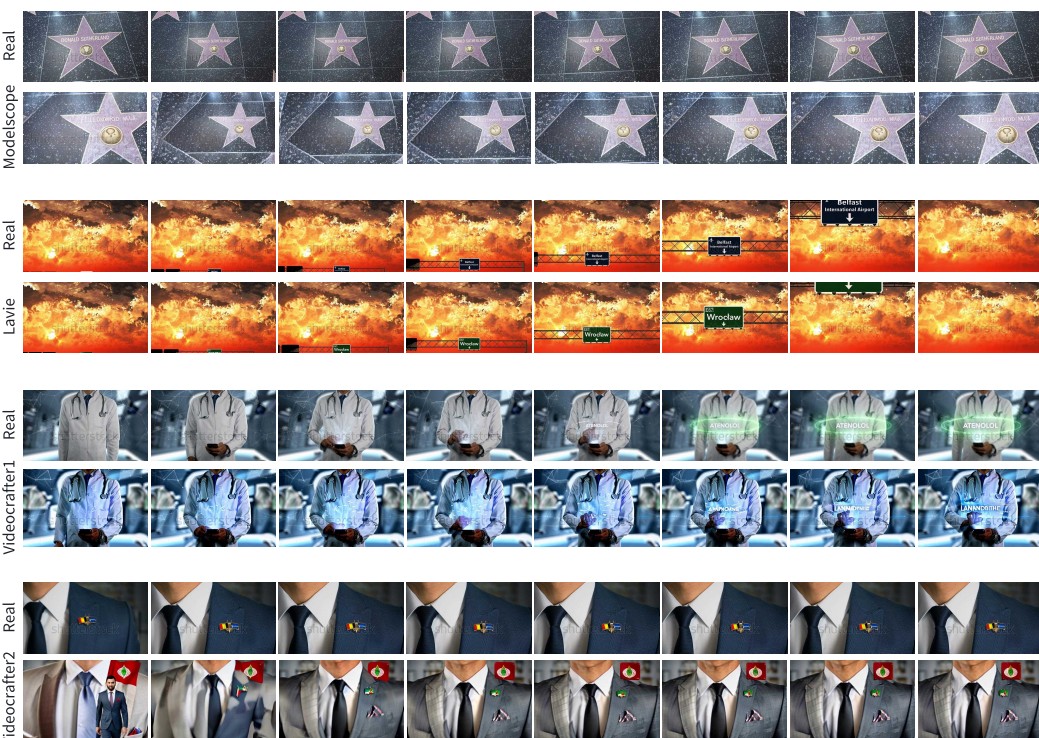

Figure 1: Video training dataset (WebVid-10M) being extracted by several open-source T2V VDMs.

of motion memorization, addressing cases where the existing video-level definition fails, especially in content-only memorization scenarios. For motion memorization, we introduce a disentangled definition that applies to all generation types, overcoming the limitations of prior definitions that are entangled with content and only applied to image-to-video (I2V) generation.

2. **Novel evaluation metrics for content and motion memorization** (Sec. 2). We develop new evaluation metrics tailored separately for content and motion memorization. Our content memorization metric shows improved accuracy over baseline, while our motion memorization metric is the first designed for VDMs. It emphasizes the detection of motion patterns that pose privacy risks and has demonstrated reasonable performance in experiments.

3. **Systematic extraction and analysis of memorization in VDMs** (Sec. 3). Using our proposed metrics and insights from IDM memorization research, we are the first to systematically extract training data from various VDMs, including text-conditional models trained on large datasets like WebVid-10M (Bain et al., 2021) and obtain valuable findings. Unlike prior research that relied on individual instances of generated videos extracted from project websites, we provide a comprehensive analysis of memorization frequency by quantifying similarity scores and memorized generation percentages using our proposed metrics. Our findings reveal that memorization is widespread, affecting not only video data but also image data used for training text-to-image (T2I) models, which serve as backbones for VDMs, introducing privacy risks. This aspect has been overlooked by previous research, which even suggests T2I backbones as a potential remedy for the issue. Moreover, we found that VDMs pretrained on public datasets and fine-tuned on larger, higher-quality private datasets still memorize public data. This suggests that the overall extent of memorization may be underestimated, as these models may also memorize private datasets.

4. **Remedies for detecting content and motion memorization** (Sec. 4). Finally, we propose remedies by adapting image memorization detection strategies for VDMs. By incorporating the temporal dimension, we develop methods capable of detecting both content and motion memorization separately with high effectiveness and efficiency. Notably, this is the first approach to enable motion memorization detection in VDMs, offering a strong foundation for future improvements in addressing memorization issues in VDMs.

## 2 DEFINING AND MEASURING VIDEO MEMORIZATION

Previous work (Rahman et al., 2024) highlights the need to analyze motion memorization in video diffusion models (VDMs) in addition to content memorization, which has been well-studied in image diffusion models (IDMs). They proposed two memorization definitions: *content and motion memorization* and *motion memorization*, along with evaluation methods. However, we identify several limitations in their definitions and metrics that may hinder broader adoption. To address this, we propose more practical definitions and more generalized, disentangled metrics for evaluating memorization in VDMs.

### 2.1 CONTENT MEMORIZATION

#### 2.1.1 RESEARCH GAP

Content memorization has been extensively studied in IDMs, where it is defined as 'reconstructive memory', which refers to the reproduction of an object that appears identically in a training image, neglecting minor variations in appearance that could result from data augmentation, with allowances for transformations like shifting, scaling, or cropping (Somepalli et al., 2023a; Chen et al., 2024a). SSCD (Pizzi et al., 2022) is the standard pre-trained model used to extract embeddings from images, and cosine similarity between SSCD embeddings quantifies content memorization.

The only work that has investigated content memorization in VDMs (Rahman et al., 2024), however, does not separately define content memorization as a disentangled definition. Instead, it combines content and motion memorization into a single definition, where a generated video is considered memorized if every frame perfectly replicates a training video in both content and motion. They correspondingly measure such *content and motion memorization* by adapting SSCD in a corresponding way, where they compute SSCD embeddings for each frame, concatenate them, and then calculate the cosine similarity of the resulting vectors, termed Video SSCD (VSSCD):

$$\text{VSSCD} = \text{cosine\_similarity} \left( \text{concat} \left( \Phi_{SSCD}(G_i) \right)_{i=1}^{N_1}, \text{concat} \left( \Phi_{SSCD}(T_j) \right)_{j=1}^{N_2} \right) \quad (1)$$

where $G_i$ represents the $i$-th frame of the generated video, which has $N_1$ frames, and $T_j$ represents the $j$-th frame of the training video, which has $N_2$ frames. The function $\Phi_{SSCD}(\cdot)$ extracts the SSCD embedding for a frame.

However, this approach has several limitations:

1. **No definition or metric for content-only memorization**. The current definition and VSSCD metric overlook cases where content is memorized, but the motion is not, narrowing the range of possible use cases and leading to incomplete evaluations.

2. **Lack of frame-level memorization detection**. Privacy concerns arise if even a single frame of a generated video memorizes a training frame. Current methods only capture video-level memorization and miss the more practical frame-level risks.

3. **No assessment of image training dataset memorization**. Many VDMs, like those built on IDMs such as Stable Diffusion, are pretrained on large-scale image datasets. This creates a risk of generated videos memorizing these image datasets, yet the current definitions and metrics only consider video-to-video comparisons. We argue for a more generalized approach that unifies memorization assessments across both video and image domains.

4. **No quantitative evaluation of the metric**. Rahman et al. (2024) only demonstrates the effectiveness of VSSCD by showing limited true positive examples with high VSSCD scores. However, it has not been evaluated using standard classification measures such as F1-score & AUC. Our analysis reveals many failure cases, motivating the need for an improved metric.

#### 2.1.2 PROPOSED DEFINITION AND METRIC

To address these limitations, we redefine content memorization as a frame-level phenomenon independent of motion memorization. A generated video is considered memorized if any single frame replicates a training video frame or a training image (in the case of single-frame data), following the rationale of 'reconstructive memory' defined in image memorization studies.

We modify the SSCD similarity measure for video memorization by calculating the frame-wise similarity for all pairs of frames between generated and training videos rather than concatenating embeddings. The maximum similarity score across all pairs is then used as the video similarity score, formalized as the Generalized SSCD (GSSCD):

$$\text{GSSCD} = \max_{1 \leq i \leq N_1,\, 1 \leq j \leq N_2} \text{SSCD}(G_i, T_j) \tag{2}$$

where $G_i$ represents the $i$-th frame of the generated video with $N_1$ frames, and $T_j$ represents the $j$-th frame of the training video with $N_2$ frames. The function $SSCD(\cdot, \cdot)$ computes the cosine similarity between the SSCD embedding of two frames. This metric generalizes to compute similarities between videos and images (when the number of frames is one), making it applicable to both domains.

### 2.1.3 EVALUATIONS

We also quantitatively evaluate the alignment of the metrics with human annotations. Since WebVid-10M serves as the training dataset for most VDMs, including ModelScope, we leveraged the most duplicated captions from WebVid-10M, which have a higher likelihood of triggering memorization, as text-conditioning for ModelScope. Using these prompts, we generated 1,000 videos, ensuring a mix of memorized and non-memorized cases. Human annotators were then asked to label these generated videos as either memorized or non-memorized, with access to the corresponding training videos and captions. These human annotations served as ground truth for computing classification metrics (AUC and F1-score) for both VSSCD and GSSCD. As shown in Tab. 1 (right), GSSCD aligns more closely with human annotations, significantly outperforming VSSCD in both AUC and F1-score. This demonstrates GSSCD's effectiveness as a metric for evaluating content memorization.

Table 1: (Left) Comparison of OFS-k with and without NMF across different values of $k$. (Right) Comparison of VSSCD and GSSCD.

| | $k = 1$ | | $k = 2$ | | $k = 3$ | | $k = 4$ | |
|---|---|---|---|---|---|---|---|---|
| | AUC | F1 | AUC | F1 | AUC | F1 | AUC | F1 |
| OFS-k w/o NMF | 0.748 | 0.627 | 0.756 | 0.633 | 0.758 | 0.633 | 0.760 | 0.640 |
| OFS-k w/ NMF | **0.954** | **0.847** | **0.968** | **0.867** | **0.974** | **0.870** | **0.962** | **0.866** |

| | AUC | F1 |
|---|---|---|
| VSSCD | 0.974 | 0.814 |
| GSSCD | **0.995** | **0.919** |

## 2.2 MOTION MEMORIZATION

### 2.2.1 RESEARCH GAP

Motion memorization is not applicable for image generations and has not been extensively explored in video diffusion models (VDMs), and has only been indirectly studied by Rahman et al. (2024). Their approach focuses narrowly on image-conditional video generation (I2V), where they define motion memorization as the inability of the model to generate novel motion patterns when conditioned on augmented versions of an initial training video frame. The rationale behind their method is that a VDM capable of understanding motion should produce consistent high-fidelity videos when conditioned on both the original and augmented initial frames (e.g., flipping, cropping).

Instead of using a direct metric to evaluate motion memorization, they propose an indirect approach: comparing the fidelity of generated videos when conditioned on the original and varied versions of the initial frame. They found that conditioning on the original frame resulted in high-fidelity outputs due to perfect memorization of both content and motion, while conditioning on variants led to artifacts attributed to the model's memorization of motion patterns from the training videos.

However, this definition and evaluation have several limitations:

1. **Limited applicability**: The method applies only to I2V scenarios and cannot be extended to other types of generations, such as unconditional or text-conditional.

2. **Ambiguity in definition and evaluation**: The reliance on fidelity as a proxy for motion memorization introduces ambiguity, as high fidelity can also result from content memorization or other factors unrelated to motion. This approach fails to isolate the contribution of memorized motion patterns and draw reliable conclusions about motion memorization.

3. **Inefficiency**: Their method is resource-intensive, requiring numerous video generations to compute the FVD for fidelity assessment. The lack of a one-to-one similarity metric between individual generation-training pairs further reduces practicality and efficiency.

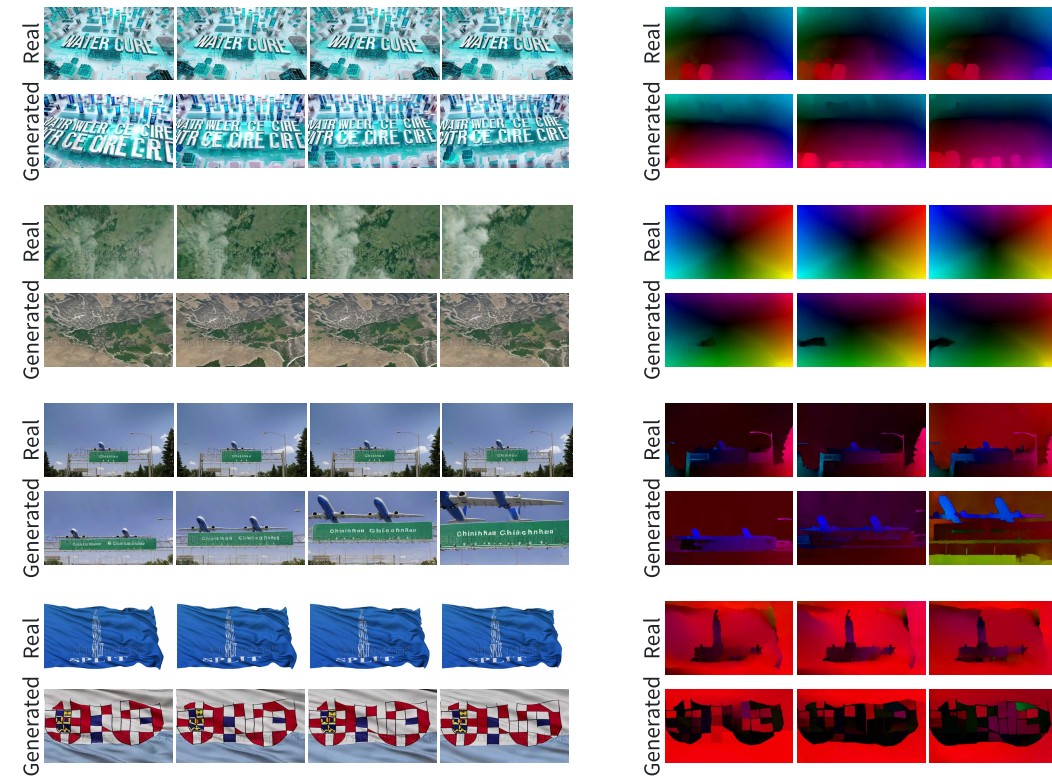

Figure 2: Motion memorization detected using OFS-3. The left side shows consecutive frame pairs from both the training video and the generated video using ModelScope, while the right side visualizes the optical flow between consecutive frames, computed using RAFT. OFS-3 calculates the average cosine similarity across three frame pairs, effectively capturing motion memorization.

### 2.2.2 PROPOSED DEFINITION AND METRIC

To address these shortcomings, we propose a more general and direct definition of motion memorization that applies to all types of video generations, not just I2V. Our approach evaluates motion alone, disentangling it from other elements like content or fidelity. We define **motion memorization** as the occurrence of high similarity in the optical flow of consecutive frames between a generated and a training video. Crucially, we disregard natural motions, such as camera panning or static frames, that pose no privacy risks. To formalize this, we introduce a novel metric called **Optical Flow Similarity (OFS-k)**. This metric quantifies the similarity between optical flows over consecutive frames, with **k** representing the number of consecutive frames considered in the computation. The key steps are:

1. **Optical flow calculation**: For each generated-training video pair, we compute the optical flow for all consecutive frames using RAFT (Teed & Deng, 2020). For each flow pair $(\mathbf{f}^g, \mathbf{f}^t)$, we calculate the **cosine similarity** between the flow vectors at corresponding pixels:

$$S_{i,j} = \frac{\sum \mathbf{f}_i^g \cdot \mathbf{f}_j^t}{\|\mathbf{f}_i^g\|\|\mathbf{f}_i^t\|} \tag{3}$$

   where $\mathbf{f}_i^g$ and $\mathbf{f}_j^t$ are the optical flow vectors for the $i$-th flow of generated video (from the $i$-th to $i+1$-th frame) and the $j$-th flow of training video (from the $j$-th to $j+1$-th frame).

2. **Max consecutive flow similarity**: For a given **k**, we compute the cosine similarity for **k** consecutive frames and define **OFS-k** as the maximum average similarity score across all **k** consecutive frame windows. The rationale for considering consecutive frames is that memorized motion often spans continuous sequences of frames, and measuring the maximum over these sequences helps capture such behavior without being overly sensitive

to isolated frame similarities. Formally, **OFS-k** is defined as:

$$\text{OFS-k} = \max_{0 \le i \le N_1 - k, \, 0 \le j \le N_2 - k} \left( \frac{1}{k} \sum_{n=0}^{k-1} S_{i+n, j+n} \right) \tag{4}$$

3. **Tuning the hyperparameter** $k$: The hyperparameter $k$ controls the length of consecutive frames considered. If $k$ is too small, the metric may become too strict, potentially resulting in a high false positive rate by categorizing natural, non-memorized motions as memorized. Conversely, setting $k$ too large could overlook cases where only a portion of the video is memorized, leading to false negatives. Thus, $k$ is a tunable parameter that can be adjusted to balance the trade-off between false positives and false negatives.

### 2.2.3 NATURAL MOTION FILTERING (NMF)

To further refine our approach, we introduce **Natural Motion Filtering (NMF)** to filter out motions that should not be classified as memorized due to natural occurrence. NMF targets two specific cases:

1. **Camera panning**: This effect involves uniform motion across all pixels in the same direction. We compute the **entropy** of the flow directions to quantify the diversity of pixel movement. Low entropy indicates uniform motion, which we consider natural and not memorized. Specifically, we compute the angle $\theta_i$ of each optical flow vector $\mathbf{f}_i$ and create a histogram of these angles. The entropy $H$ is then calculated as:

$$H = -\sum p(\theta_i) \log p(\theta_i) \tag{5}$$

where $p(\theta_i)$ is the probability of a flow vector having direction $\theta_i$. If $H$ falls below a threshold, the motion is classified as camera panning and is ignored.

2. **Static frames**: Frames with minimal movement are considered static. We calculate the average magnitude of the optical flow vectors, and if the magnitude is below a specified threshold, the frame is categorized as static and excluded from further analysis.

By applying NMF, we ensure that only meaningful, privacy-sensitive motion patterns are evaluated as potential memorized motions, filtering out natural and non-threatening movements.

### 2.2.4 EVALUATIONS

We also quantitatively evaluated the alignment of our proposed OFS-k metric with human annotations, following the same procedure used for comparing content memorization metrics. As shown in Tab. 1 (left), OFS-k achieves high classification performance for motion memorization, providing a valuable tool for further research in this area. Additionally, we performed an ablation study on the NMF and hyperparameter analysis for $k$ in OFS-k. From Tab. 1 (left), we observe that incorporating NMF significantly improves AUC and F1-score, demonstrating its effectiveness in filtering out similar but non-memorized natural motions. This improvement aligns with the more practical notion of motion memorization, emphasizing the importance of privacy preservation. Among the tested values of $k$, we found that an intermediate value of $k = 3$ achieves the best trade-off, yielding the highest AUC and F1-score, whereas $k = 2$ achieves the second-best. This supports our earlier intuition that setting $k$ too low makes the metric overly strict, potentially leading to a high false positive rate by categorizing natural, non-memorized motions as memorized. On the other hand, setting $k$ too high risks missing cases where only part of the video is memorized, resulting in false negatives.

We also present qualitative results in Fig. 2 and 9, where Fig. 2 highlights examples of video pairs with high OFS-3 scores. In the first and second rows, the motions are characterized by a unique and different rotational pattern, with some regions remaining static while others show varying flow magnitudes. These pair reports OFS-3 scores of 0.9012 and 0.9393. In the third row, the plane flies over a horizontal road sign with a similar motion, yielding a score of 0.8523. The final row depicts the motion of a flag, where the logo remains static while the surrounding area flows in a highly similar manner, reporting a score of 0.8806. In Fig. 9, we present examples of training and generated video pairs with similar optical flow but representing natural motions like camera panning that do not pose privacy risks. In these cases, the optical flow vectors for each pixel exhibit uniform angles, reflecting consistent motion across the frames. Our proposed NMF effectively filters out such natural motions, ensuring they are not counted as memorized cases.

## 3 MEMORIZATION IN VIDEO DIFFUSION MODELS (VDMs)

Previous work (Rahman et al., 2024) only investigated generated videos sourced from project websites of several VDMs and identified instances of their memorization of the training data. However, the frequency and extent of these memorization issues remain unclear. To address this, we leverage insights into the causes of memorization in diffusion models and apply our proposed metrics for quantifying memorization. This allows us to systematically extract memorized cases and report statistics on the degree and frequency of memorization across multiple VDMs.

In terms of the scope, previous work focused only on memorization issues in unconditional and image-conditional VDMs using small datasets such as UCF-101 (Soomro et al., 2012) and Kinetics-600 (Carreira & Zisserman, 2018). However, for text-to-video (T2V) generation, their study lacked a comprehensive investigation of various VDMs trained on larger datasets like WebVid-10M (Bain et al., 2021), which serves as the foundation for many VDMs used in video generation (Singer et al., 2022; Wang et al., 2023a; Chen et al., 2023) and video editing (Guo et al., 2023a;b; Zhang et al., 2024). To address this gap, we analyzed the WebVid-10M dataset and conducted the first investigation into the memorization behavior of several open-source VDMs (Wang et al., 2023a;b; Chen et al., 2023; 2024b) trained on this dataset.

Additionally, their scope only addresses the VDMs' memorization of video training data, while neglecting the memorization of image training data. Specifically, Rahman et al. (2024) reported that VDMs extended from pretrained text-to-image (T2I) models can benefit from their inherent capability to generate creative image outputs, which can reduce memorization on video datasets. However, we have demonstrated that this benefit comes at the cost of increased memorization of the image data inherent in the pretrained T2I backbone. This reveals an overlooked trade-off where reducing video dataset memorization leads to the unintended consequence of memorizing image data.

### 3.1 EXTRACTING TRAINING DATA FROM VIDEO DIFFUSION MODELS

Previous studies on memorization in image diffusion models (IDMs)(Carlini et al., 2023; Somepalli et al., 2023b; Webster, 2023) have shown that a significant cause of memorization is the duplication of training data, which leads to overfitting. Leveraging these insights, we begin our analysis by examining the duplication within the widely used WebVid-10M dataset(Bain et al., 2021). We extract features from the dataset using the Inflated 3D Convolutional Network (I3D)(Carreira & Zisserman, 2017), a widely used video feature extractor pretrained on Kinetics(Carreira & Zisserman, 2018). I3D extends 2D convolutional networks to 3D, enabling the capture of spatiotemporal features. We compute cosine similarities between training data pairs and identify the most duplicated videos. From this analysis, we extracted the first 500 unique captions corresponding to the most duplicated videos, forming the WebVid-duplication-prompt dataset, which will be used for extracting memorization cases in various video diffusion models (VDMs).

For analyzing VDMs' memorization of image datasets, we utilize the open-sourced prompt datasets provided by Webster (2023). These datasets organize the most commonly memorized prompts for several IDMs, including Stable Diffusion's memorization of the LAION dataset.

Table 2: Quantifying the percentage of memorized generations and average similarity scores for both content and motion memorization across various open-source VDMs, evaluated on both video and image datasets.

| | Content Memorization | | Motion Memorization | |
|---|---|---|---|---|
| | %Mem | Similarity | %Mem | Similarity |
| T2V - ModelScope (WebVid-10M) | 18.02 | 0.33 | 33.10 | 0.31 |
| T2V - ModelScope (LAION) | 15.40 | 0.29 | - | - |
| T2V - LaVie (WebVid-10M) | 20.21 | 0.33 | 25.25 | 0.26 |
| T2V - VideoCrafter1 (WebVid-10M) | 3.81 | 0.28 | 19.25 | 0.19 |
| T2V - VideoCrafter2 (WebVid-10M) | 3.93 | 0.27 | 22.68 | 0.19 |
| UV - RaMViD (UCF-101) | 45.00 | 0.43 | 23.75 | 0.09 |

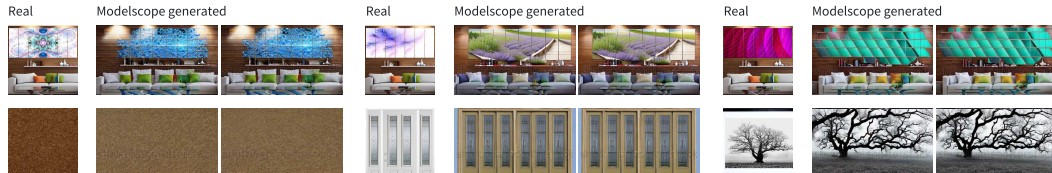

Figure 3: Image training dataset (LAION) being extracted by ModelScopeT2V.

## 3.2 ANALYSIS

**Text-conditional generations**. To evaluate text-conditional (T2V) models' memorization of video training data, we conducted experiments on several open-source VDMs trained or partially trained on WebVid-10M, including ModelScope (Wang et al., 2023a), LaVie (Wang et al., 2023b), VideoCrafter1 (Chen et al., 2023), and VideoCrafter2 (Chen et al., 2024b). As shown in Tab. 2 and Fig. 1, memorization is widespread across all models using WebVid-10M as part of their training set. Notably, LaVie also utilized the closed-source Vimeo25M dataset, which contains 25 million high-definition, watermark-free text-video pairs, alongside WebVid-10M during training. Intuitively, this should allow the model to generalize better by learning from more diverse sources. However, it still exhibits a 20.21% memorization rate on WebVid-10M. Due to the closed-source nature of the Vimeo25M dataset, we are unable to investigate its memorization extent, but we believe similar risks apply. This highlights the need for researchers to ensure that even closed-source datasets are free of copyrighted images and videos, as they may pose as legitimate risks as the open-source ones. Similarly, VideoCrafter1 and VideoCrafter2 also report training on private datasets in addition to WebVid-10M. Therefore, the numbers reported in Tab. 2 may underestimate the severity of the memorization issue, as the risk of memorizing content from other datasets remains unaccounted for.

For analyzing the memorization of image training data in T2V models, we experimented with open-source VDMs (Wang et al., 2023a) that utilize text-to-image (T2I) model backbone of Stable Diffusion (Rombach et al., 2022). From Tab. 2 and Fig. 3, we observe evidence of ModelScope's generations replicating the image training data. This demonstrates that VDMs are also capable of memorizing image training data, a phenomenon overlooked in the current literature.

**Unconditional generations**. For evaluating unconditional (UV) models, we analyzed RaMViD (Höppe et al., 2022) pretrained on UCF-101 (Soomro et al., 2012). We found that 45.00% of 1,000 generations were memorized instances. As RaMViD was not pretrained on any T2V models, we did not analyze its memorization of image datasets. Qualitative examples can be found in Fig. 4.

## 4 POTENTIAL REMEDIES

### 4.1 MOTIVATION

The only previously proposed solution for mitigating memorization in VDMs involves using a text-to-image diffusion model as a backbone, then fine-tuning it on a video dataset to reduce memorization of video data (Rahman et al., 2024). However, as shown in Sec. 3, this approach leads to the unintended consequence of the model memorizing image data, thus reducing the overall effectiveness of this strategy.

As a preliminary investigation into memorization in VDMs, we draw inspiration from the IDM memorization domain, which offers several potential strategies. One such approach is to retrain VDMs on de-duplicated datasets, as duplication has been identified as a major factor leading to overfitting and memorization. This suggests that future pre-training of VDMs should prioritize training on de-duplicated data or using pre-trained backbones that have been trained on such data. However, two challenges arise: First, duplication is not the only cause of memorization. For example, text captions in different expressions may share similar meanings, which could still trigger memorization when the user prompt is overly specific. Second, retraining on de-duplicated data is computationally expensive and inefficient. Another approach involves adapting mitigation strategies from the image domain to VDMs, such as adding randomness to input prompts (Somepalli et al., 2023b), employing prompt engineering (Wen et al., 2024), or leveraging guidance strategies during

inference (Chen et al., 2024a). However, these strategies typically come at the cost of utility, such as lower quality or reduced text alignment in the generated outputs.

In contrast, we argue that inference-time detection strategies are more practical and impactful. These strategies can effectively and efficiently reduce memorized generations by allowing for early exits during inference upon detecting memorization. In terms of effectiveness, inference-time detection strategies do not modify the inference process itself but rely on memory-efficient cache signals to detect memorization, thereby fully preserving output utility. In terms of efficiency, these strategies avoid the computational infeasibility of searching through entire training datasets for detection, which is challenging for high-performing models trained on vast amounts of data. Instead, inference-time detection strategies can perform detection during the inference process, which typically takes only tens of seconds. Moreover, our proposed detection method can achieve accurate detection within 1-step and 10-step inference processes that only take seconds.

Thus, developing an efficient inference-time detection strategy for both types of memorization in VDMs could significantly contribute to the field. To this end, we adapt the image memorization detection strategy (Wen et al., 2024) to the video domain, incorporating the temporal dimension to enable the detection of both content and motion memorization. This adaptation serves as a solid foundation for future improvements.

## 4.2 DETECTION STRATEGIES FOR BOTH CONTENT AND MOTION MEMORIZATION

In the IDM memorization domain, a recent work (Wen et al., 2024) introduces a method for memorization detection using the magnitude of text-conditional predictions, expressed as:

$$m_t = \|\epsilon_\theta(x_t, e_p) - \epsilon_\theta(x_t, e_\phi)\|_2 \tag{6}$$

This method builds on the observation that, under identical initialization, outputs conditioned on different text prompts tend to exhibit similar semantic properties, resulting in relatively small magnitude values throughout the inference process. However, for prompts $e_p$ that are prone to memorization, the text condition $\epsilon_\theta(x_t, e_p)$ consistently steers the generation toward memorized outcomes, regardless of initialization, thereby producing significantly larger magnitude values. Thus, a larger magnitude serves as a signal for potential memorization.

We extend this detection strategy to VDMs to enable both content and motion memorization detection, adapting it to account for the temporal dimension inherent in videos. In IDM, the magnitude is a 3-dimensional tensor with the shape $[C, H, W]$, where $C$ is the number of channels, and $H$ and $W$ represent the height and width of the latent noise prediction $\epsilon_\theta(x_t, e_p)$ and $\epsilon_\theta(x_t, e_\phi)$. In VDMs, this expands to a 4-dimensional tensor with the shape $[C, F, H, W]$, where $F$ is the number of frames in the video.

### 4.2.1 CONTENT MAGNITUDE

For content magnitude, we follow the same intuition behind our content memorization metric (GSSCD) that emphasizes frame-wise memorization, where frame-wise similarities are computed, and the maximum value is used as the overall score. Similarly, we compute the frame-wise magnitude for each of the $F$ frames, resulting in $F$ magnitudes. The maximum magnitude is then taken as the content magnitude:

$$m_{\text{content}} = \max\{\|\epsilon_\theta(x_t^i, e_p) - \epsilon_\theta(x_t^i, e_\phi)\|_2 : i = 1, \dots, F\} \tag{7}$$

This maintains coherence and simplicity while also resulting in effective detecting content memorization.

### 4.2.2 MOTION MAGNITUDE

For motion magnitude, our approach leverages the temporal dimension by analyzing frame transitions. The frame transition is defined as the difference between each consecutive pair of frames. Given $F$ frames, this results in $F - 1$ transitions. To compute the motion magnitude, we compare the transitions between the text-conditional prediction and the unconditional prediction:

$$m_{\text{motion}} = \max\{\|\Delta\epsilon_\theta(x_t^i, e_p) - \Delta\epsilon_\theta(x_t^i, e_\phi)\|_2 : i = 1, \ldots, F-1\} \tag{8}$$

where $\Delta\epsilon_\theta(x_t^i, e_p) = \epsilon_\theta(x_t^{i+1}, e_p) - \epsilon_\theta(x_t^i, e_p)$ represents the frame transition at step $i$. This method captures whether the transitions between frames in the text-conditional generation are "abnormal" compared to the unconditional prediction, thus signaling motion memorization that overfits the motion in the training videos.

**Remark on the alternative approach of using optical flow**. We also explored using optical flow to quantify motion transitions. This involved decoding intermediate results of $\epsilon_\theta(x_t^i, e_p)$ and $\epsilon_\theta(x_t^{i+1}, e_p)$ pairs for each frame $i$ from each inference step $t$ back into image space using the diffusion model's decoder, followed by computing optical flow with RAFT (Teed & Deng, 2020). However, this approach yielded unsatisfactory results, as the decoded images were likely out-of-distribution for RAFT compared to the dataset it was trained on. This mismatch led to inaccurate flow estimations, making this method inferior to our proposed approach. Another limitation of using RAFT in this case is the significant additional computational cost, as the RAFT model would need to be invoked multiple times during every inference step. By directly leveraging the latent space through frame transitions, our motion magnitude method remains simple yet effective for detecting motion memorization.

Table 3: Comparison of content and motion memorization detection performance at different inference steps, using content and motion magnitude as a signal respectively.

|          | Content Memorization | | | Motion Memorization | | |
|----------|-------|----------|----------|-------|----------|----------|
|          | AUC   | F1-score | Time (s) | AUC   | F1-score | Time (s) |
| 1-step   | 0.892 | 0.749    | **0.709**| 0.814 | 0.614    | **0.709**|
| 10-step  | 0.895 | 0.896    | 7.094    | 0.926 | 0.775    | 7.094    |
| all-step | **0.978** | **0.904** | 35.470 | **0.933** | **0.837** | 35.470 |

### 4.2.3 PERFORMANCE EVALUATION

Tab. 3 presents a comparative analysis of content and motion memorization detection performance across different inference step strategies, using AUC, F1-score, and inference time as evaluation metrics. The 1-step approach leverages the magnitude computed at the first inference step, offering the advantage of rapid detection with a processing time of just 0.709 seconds while already achieving decent performance. As the number of inference steps increases, both the 10-step and all-step approaches demonstrate significant improvements in detection accuracy. The all-step method, which uses the average magnitude over all 50 inference steps, achieves the highest performance with AUC scores of 0.978 for content memorization and 0.933 for motion memorization, and F1-scores of 0.904 and 0.837, respectively. The results suggest a trade-off between computational efficiency and detection performance. We argue that the 10-step approach balances efficiency and accuracy, offering notable improvements in detection performance without the full computational burden of the all-step method.

## 5 CONCLUSION

In this paper, we address the critical issue of memorization in video diffusion models (VDMs), a largely overlooked area compared to image diffusion models (IDMs). We introduced new, privacy-focused definitions for both content and motion memorization, as well as tailored evaluation metrics that separately assess these types of memorization in VDMs. Armed with such metrics, we are the first to systematically extract training data from large video and image datasets and report quantitative results on the extent of memorization in various open-source VDMs. Our findings demonstrate that memorization is a widespread issue, not only in video datasets but also in image datasets used to train text-to-image (T2I) backbones, which has been previously underestimated in the literature. Furthermore, we propose effective remedies by adapting image memorization detection strategies to the video domain. These strategies, incorporating the temporal dimension, offer efficient detection of both content and motion memorization, providing a robust foundation for future improvements in privacy preservation within VDMs, particularly as their usage continues to grow.

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

# A  APPENDIX

## A.1  RELATED WORK

**Video diffusion models**. Video diffusion models (VDMs) extend the diffusion framework, initially developed for image generation, to model the temporal dynamics in videos. Building on the success of image diffusion models (IDMs) such as Denoising Diffusion Probabilistic Models (DDPMs)(Ho et al., 2020; Nichol & Dhariwal, 2021) and Stable Diffusion(Rombach et al., 2022), VDMs tackle the added complexity of maintaining temporal consistency across frames. Early works, like Video Diffusion Models (Ho et al., 2022), adapted the diffusion process by incorporating 3D convolutional architectures to handle both spatial and temporal dimensions. Recent VDM advancements such as Make-A-Video (Singer et al., 2022), which leverage pretrained text-to-image (T2I) backbones like Stable Diffusion (Rombach et al., 2022), have pushed the field forward, expanding the application of VDMs to conditioned video generation from text prompts. Other models, such as Runway Gen-1 (Esser et al., 2023), also push boundaries by incorporating text, images, and videos as conditional inputs to better control generated content. The popularity of VDMs has further surged with advancements such as Sora (OpenAI, 2024), Gen-3 (Runway, 2024), and discord-based servers (Discord, 2023), which have garnered significant attention on social media. Despite this increasing interest, the phenomenon of memorization in VDMs remains largely underexplored.

**Memorization in diffusion models**. Memorization in image diffusion models (IDMs) has been extensively studied. Pioneer works such as Carlini et al. (2023) and Somepalli et al. (2023a) successfully extracted training data from IDMs, including DDPMs and Stable Diffusion, using datasets like CIFAR-10 (Krizhevsky, 2009) and LAION (Schuhmann et al., 2022). This motivates the subsequent works (Somepalli et al., 2023b; Daras et al., 2023; Gu et al., 2023; Wen et al., 2024; Chen et al., 2024a; Ren et al., 2024) to work on analyzing the causes and proposing remedies. Recently, Rahman et al. (2024) explored memorization in VDMs, but the study's definitions and evaluations of content and motion memorization are limited in scope and lack broad applicability. Additionally, their analysis was based on individual instances of generated videos extracted from project websites, rather than a systematic study of training data and the extent of memorization. Furthermore, the study did not explore memorization in text-to-video (T2V) models trained on widely used datasets like WebVid-10M.

In this work, we address these limitations by conducting a thorough investigation on several open-source VDMs, quantifying the percentage of memorized generations and average similarity scores for both content and motion memorization across both video and image datasets.

## A.2  ADDITIONAL QUALITATIVE RESULTS

We present the following additional qualitative results:

- Fig. 4 present results of video training dataset (UCF-101) being extracted by RaMViD's unconditional generation.

- Fig. 5, 6, 7, 8 present additional results of video training dataset (WebVid-10M) being extracted by several open-source T2V VDMs.
- Fig. 9 present examples of training and generated video pairs with similar optical flow but representing natural motions like camera panning that do not pose privacy risks.

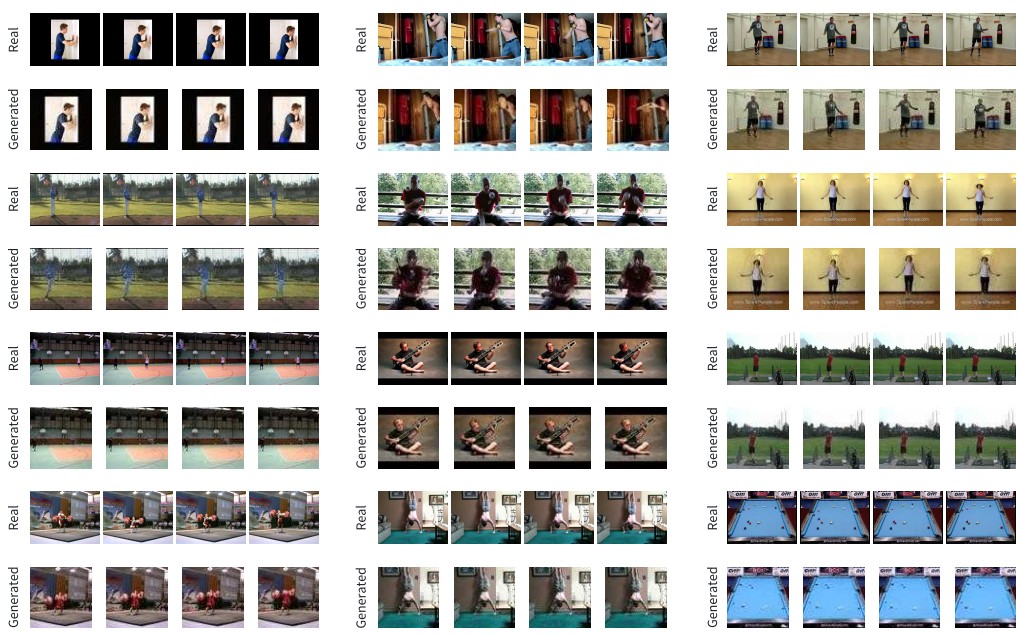

Figure 4: Video training dataset (UCF-101) being extracted by RaMViD's unconditional generation.

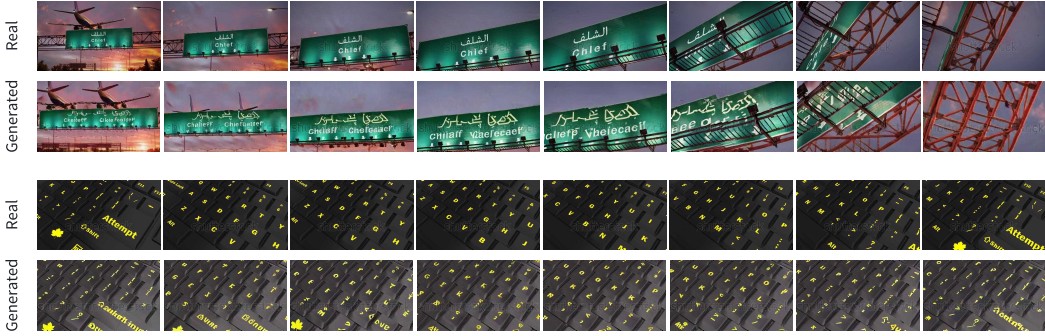

Figure 5: Video training dataset (WebVid-10M) being extracted by ModelScope.

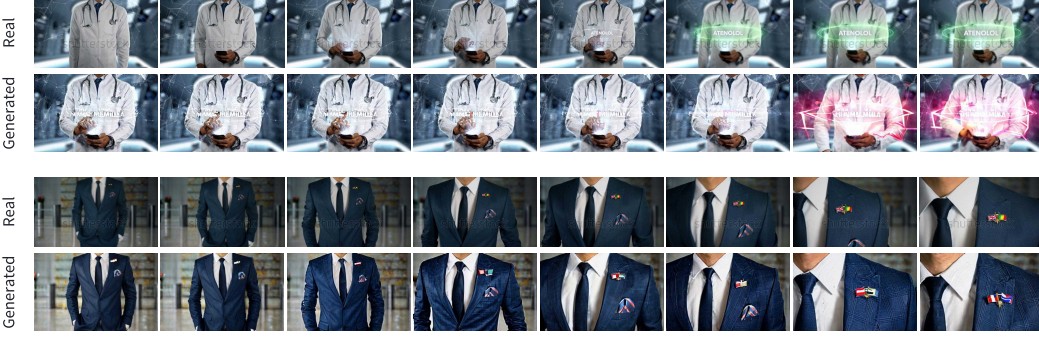

Figure 6: Video training dataset (WebVid-10M) being extracted by LaVie

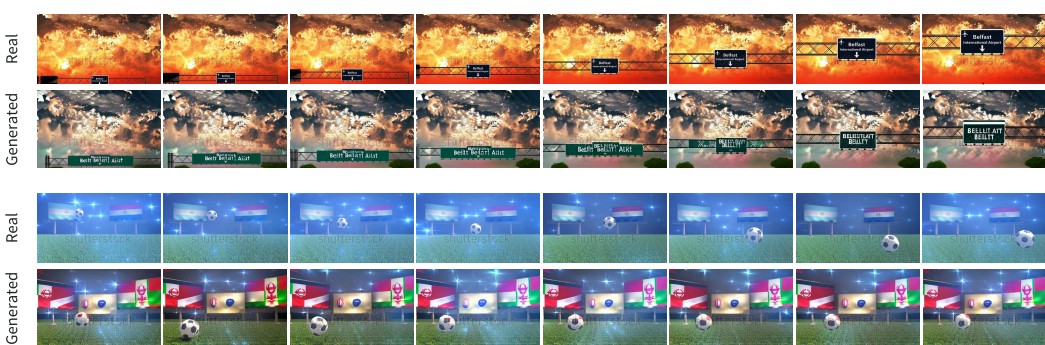

Figure 7: Video training dataset (WebVid-10M) being extracted by VideoCrafter1

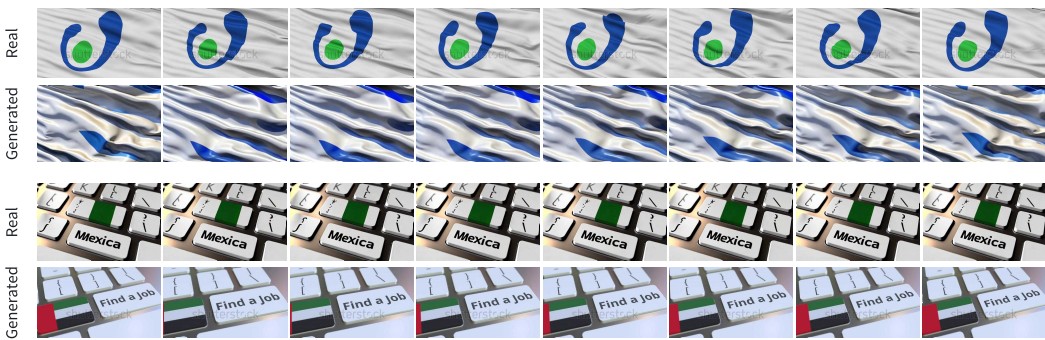

Figure 8: Video training dataset (WebVid-10M) being extracted by VideoCrafter2

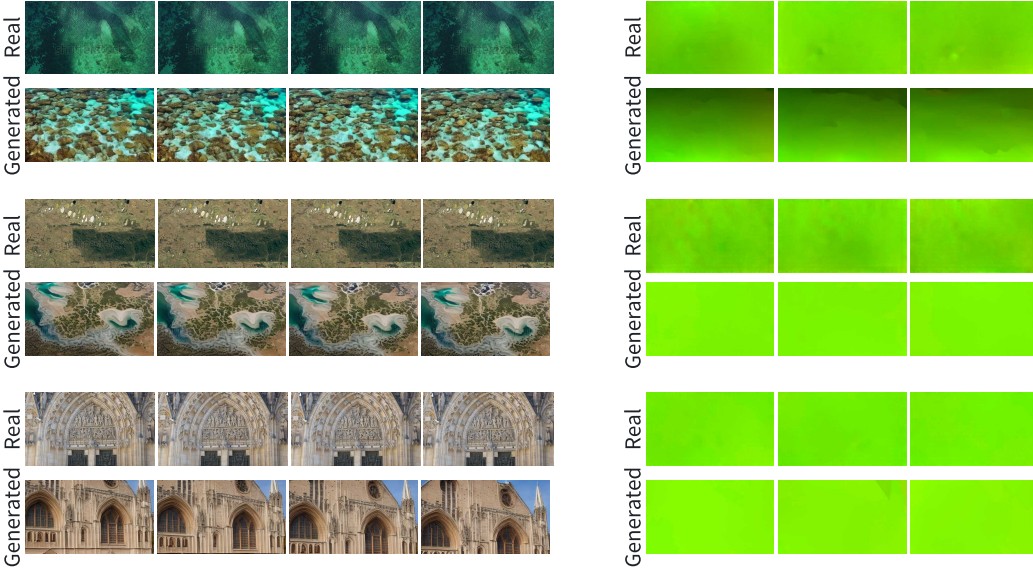

Figure 9: Examples of training and generated video pairs with similar optical flow but representing natural motions like camera panning that do not pose privacy risks. In these cases, the optical flow vectors for each pixel exhibit uniform angles, reflecting consistent motion across the frames. Our proposed NMF effectively filters out such natural motions, ensuring they are not counted as memorized cases. The left side shows consecutive frame pairs from both the training video and the generated video using ModelScope, while the right side visualizes the optical flow between consecutive frames, computed using RAFT.

