# OpenReview forum: "Investigating Memorization in Video Diffusion Models"
_ICLR.cc/2025/Conference — ICLR 2025 Conference Withdrawn Submission_

### Official Review · Reviewer_Lsij · 2024-10-28

**Soundness:** 1
**Presentation:** 1
**Contribution:** 1
**Rating:** 3
**Confidence:** 5

**Summary:**

Memorization/Replication is an important concern in current diffusion models. The paper discusses this concern from the perspective of video diffusion models. Specifically, it redefines some concepts, evaluates video diffusion model memorization, and proposes solutions.

**Strengths:**

NA

**Weaknesses:**

I have some concerns:
1. The proposed GSSCD is just a naive extension of SSCD. For video copy detection (an area you may know) research, this aggregation (GSSCD) is very common. Please see the recent VSCD competition by Meta AI. In addition, SSCD is not a good metric itself for copy detection in the background of the diffusion model (Do you know why?). You should design something to substitute SSCD instead.
2. The solutions and evaluations are also naive :) I want to see some insightful methods.
3. The video diffusion models used in the paper seem a little old, for instance, you may use the SOTA like Open-Sora, CogVideo, and etc. The poor performance of the used video diffusion models prevent you from drawing useful conclusions.

**Questions:**

In conclusion, from my point of view (who is specialized in memorization/copy detection; replication in video/image diffusion models), this paper does not contribute significantly to the field. And I am pretty sure of my judgment. No rebuttal is expected.

---

### Official Review · Reviewer_bX8J · 2024-10-30

**Soundness:** 2
**Presentation:** 2
**Contribution:** 3
**Rating:** 5
**Confidence:** 4

**Summary:**

This paper addresses the important issue of memorization in video diffusion models (VDMs) and introduces new definitions for both content and motion memorization. The study observes that memorization is a widespread problem not only in video datasets but also in image datasets. Furthermore, it proposes effective adaptations of image memorization detection methods for the video domain. In summary, the paper rethinks content and motion memorization and provides a systematic analysis of the memorization phenomenon in VDMs.

**Strengths:**

1. This paper redefines content and motion memorization, achieving better alignment with human perception.
2. It proposes remedies for detecting content and motion memorization, which are crucial for mitigating privacy risks.
3. The experimental results are comprehensive and reliable.

**Weaknesses:**

1. The writing needs improvement. In my opinion, the paper reads more like an experimental report. The contribution section is overly long, and there are frequent direct comparisons with previous work throughout the paper. These comparisons are often presented in bullet points, with each point being lengthy. Additionally, in Table 1, what do the red text and bold font represent? This is not explained in the paper.
2. The paper lacks coherent expression, making it difficult to read smoothly. Each section feels disconnected from the others, affecting the overall flow.
3. In Table 1, the notations OFS-k and NMF appear for the first time, but there is no explanation provided in the table’s caption, leaving readers confused when interpreting the content.

**Questions:**

1. Can the methods proposed in this paper be applied to known video diffusion transformers such as CogVideoX and similar models?
2. In Section 2.2.2, why is the optical flow between consecutive frames used to detect motion memorization? Could you provide some examples or additional insights to help clarify this concept?

---

### Official Review · Reviewer_MtHT · 2024-11-02

**Soundness:** 2
**Presentation:** 2
**Contribution:** 3
**Rating:** 5
**Confidence:** 3

**Summary:**

This paper investigates the memorization issue in video diffusion models. Compared to prior work, this paper introduces measures for frame-level and content memorization in generated videos. It also examines memorization from image training data and adapts image memorization detection strategies as solutions for video diffusion models.

**Strengths:**

1. This paper provides valuable insights by identifying several shortcomings in prior studies investigating the memorization of video diffusion models.
2. It proposes measuring frame-level memorization in video diffusion models, extending beyond the video-level memorization focus of previous work.
3. This paper also measures memorization from the image training dataset, which is beneficial.

**Weaknesses:**

This paper points out several problems of prior study in investigating the memorization problem of video diffusion models, which are valuable. However, there are the following concerns that make the paper less solid and convincing.

1. It is doubtful that content and motion memorization can be disentangled as in this paper. This paper uses optical flow to measure motion memorization, as indicated in Section 2.2. However, optical flow is also influenced by the content, as shown in examples with similar content in Figure 2 (like the first and third examples).

2. Memorization detection methods detect but do not solve the memorization problem. For instance, if a user wants to generate a video based on a text prompt, but the generated video is detected to have memorization issues, what should the system do to solve the memorization issue while meeting the user's requirement? Discard the video and regenerate based on the prompt?

3. Clarification is needed on metrics computation in Table 2. It would be helpful to provide specific equations or pseudocode to show how these metrics are calculated.

4. The paper organization is sometimes confusing. For example, Table 1 (left) on page 4 is first referenced on page 6, after the reference of Table 1 (right) on page 6.  Additionally,  table titles lack clarity, making it difficult for readers to understand the key conclusions from tables.

**Questions:**

Please refer to the weaknesses mentioned above.

---

### Official Review · Reviewer_mwvA · 2024-11-02

**Soundness:** 2
**Presentation:** 3
**Contribution:** 2
**Rating:** 3
**Confidence:** 2

**Summary:**

This research explores the problem of memorization in video diffusion models (VDMs), extending the concerns previously identified in image diffusion models (IDMs). This memorization can result in the generation of copyrighted content, posing legal and ethical challenges. To tackle this, the paper extends Rahman et al., 2024 by improving definitions and metrics to quantify both content and motion memorization in VDMs, referred to as GSSCD and OFS-k respectively. The authors provide a memorization analysis across various VDMs and propose an innovative detection method that leverages these newly introduced metrics, GSSCD and OFS-k, to effectively identify memorization videos. These contributions are aimed at mitigating the risks associated with memorization in video generation, enhancing the trustworthiness and applicability of VDMs.

**Strengths:**

1.	The investigation into memorization in video diffusion models (VDMs) addresses a critical and timely concern. As VDMs become increasingly sophisticated and widely used in generating realistic video content, the potential for these models to inadvertently memorize and replicate training data poses significant legal and ethical risks.

2.	The paper enhances the conceptual framework and analytical tools available for studying memorization in VDMs by introducing more nuanced and practical definitions of content and motion memorization.

3.	The paper proposes practical strategy for detecting memorization.

**Weaknesses:**

1.	The paper introduces a new metric for evaluating content memorization in video diffusion models in a frame-level manner but does not sufficiently justify the need for these over existing methods developed for image diffusion models. It remains unclear whether the current metrics for IDMs are insufficient for assessing memorization in videos.

2.	Limited Validation: The new metrics' validation primarily relies on 1K manual annotations on a single dataset (WebVid-10M), which might not reflect the diversity of real-world scenarios. More extensive validation across diverse datasets and more automated, objective evaluation methods could strengthen the findings.

3.	The paper repeatedly mentions the existence of a method in the field that addresses motion memorization in video diffusion models (Rahman et al., 2024), yet it fails to provide a qualitative comparison with this method. The absence of a direct, detailed comparative analysis weakens the argument for the proposed method's novelty and effectiveness.

4.	Additional minor weaknesses:

a.	Figure 3 appears to be redundant, as Table 2 already sufficiently demonstrates that ModelScope’s generations replicate the image training data.

b.	In Table 2, it would improve clarity if text-conditional generations (T2V) and unconditional generations (UV) were distinguished in separate column.

**Questions:**

Please refer to the previously noted weaknesses, and additionally, consider the following questions:

1.	Could the authors discuss any limitations of their work? Including such details would provide a more balanced view and help readers understand the potential constraints or challenges in applying the proposed methods in different contexts.

2.	How does the method perform under varied conditions, such as different video qualities or motion complexities? Is there a risk of the detection methods being too sensitive or not sensitive enough?

3.	Given that the authors have demonstrated the separate detection of content and motion memorization, could combining these methods into an ensemble approach enhance the overall detection performance

4.	Will you provide access to the implementation (code) or datasets used in the experiments? Providing these resources could greatly enhance the credibility and applicability of the research.

---

### Note · Authors · 2024-11-15

I have read and agree with the venue's withdrawal policy on behalf of myself and my co-authors.